# Increasing Heart Rate Variability through Progressive Muscle Relaxation and Breathing: A 77-Day Pilot Study with Daily Ambulatory Assessment

**DOI:** 10.3390/ijerph182111357

**Published:** 2021-10-28

**Authors:** Daniel Groß, Carl-Walter Kohlmann

**Affiliations:** Department of Educational and Health Psychology, University of Education Schwäbisch Gmünd, Oberbettringerstraße 200, 73525 Schwäbisch Gmünd, Germany; carl-walter.kohlmann@ph-gmuend.de

**Keywords:** heart rate variability, ambulatory assessment, resonance frequency training, progressive muscle relaxation, healthy individuals

## Abstract

The aim of this study was to examine whether it is possible to gradually increase heart rate variability (HRV) in healthy individuals (21 participants, *M* = 21.24 years, *SD* = 1.57, range 19 to 26) through regular exercises of average resonance frequency training (RFT; 6 breaths/min; 5 min each day) and progressive muscle relaxation (PMR; three times a week for 20 min). The effects were tested against an active control group using a linear mixed effect model with random slopes (day), random intercepts (participants) and an autoregressive error term. The special feature of this pilot study is that HRV was measured every day in an ambulatory assessment over 77 days, so that graduate long-term effects on HRV can be mapped. The results indicated that the PMR group significantly increased their HRV compared to the active control group. However, no effect was observed for the RFT group. Possible explanations for these results and important recommendations for subsequent studies are provided.

## 1. Introduction

Increased vagally mediated heart rate variability (HRV) is in general associated with better self-regulation at the emotional, cognitive, social, and health domains [1,2,3,4]. HRV results from the parasympathetic activity of the autonomic nervous system (ANS), in particular of the vagus nerve (also, via the stellate ganglia), influencing the sinoatrial node (a small muscle strip in the upper part of the right atrium that is sympathetic influenced by the neurotransmitter of norepinephrine and parasympathetic influenced by the neurotransmitter of acetylcholine), the primary pacemaker of the heart leading to a reduced heart rate (HR) and increased interbeat intervals (IBIs) between two consecutive heartbeats (also called RR orNN intervals) [4,5].

### 1.1. Resonance Frequency Training

It is often demonstrated in clinical populations (however, in this population HRV is often greatly reduced), that HRV can be increased through slow breathing, but also some studies indicated HRV increase for healthy individuals in longitudinal studies using a pre- and post-measurement design [6,7,8]. A slow breathing method which focus on HRV maximization is average resonance frequency training (RFT; for a systematic and meta-analytic review, see [1]). It focuses on teaching the individual to breathe at 0.1 Hertz (Hz; 6 breaths/min, i.e., 5 s inhalation and 5 s exhalation), the point where breathing and HR oscillate generally at the same resonance frequency [6,9,10,11,12]. HR oscillations do not normally occur at the same frequency as respiration. Generally, HR increases during inhalation since the cardiovascular center inhibits vagal tone influence and decreases during exhalation since the cardiovascular center restores vagal outflow via the release of acetylcholine (this phenomenon is also known as respiratory sinus arrhythmia) [3,13]. HR and respiration only oscillate at the same resonance frequency when the normal breathing frequency, usually between 0.15 and 0.4 Hz (9 and 24 breaths/minute), is rhythmically stimulated by paced breathing at an average frequency of 0.1 Hz. In general, breathing at 0.1 Hz also exercises the baroreflex to enhance context-specific regulation of blood pressure, and leads to the most efficient gas exchange and produces meditation- or mindfulness-related effects by focusing the attention on one’s breathing [6,9,10,13] (for overviews of the psychophysiological mechanisms and effects of slow breathing techniques see [10,14,15]) Several RFT studies have demonstrated that breathing at a rate of 0.1 Hz clinically improves symptoms of physical and mental disorders (for a summary, see [6]) including asthma [16], Posttraumatic Stress Disorder [17,18], and depression [19,20,21]. Most clinical studies exist for anxiety and cardiovascular disorders, where there is strong evidence of a small to medium effect of RFT [1]. However, only a small number of studies have investigated the effect of RFT on HRV among healthy individuals in cross-sectional or pre- and post-measurement designs, and the effects found are sometimes inconsistent. Siepmann and colleagues [20] found that a 6-session RFT (duration of one session was 25 min) did not increase HRV among healthy individuals, whereas Lehrer and colleagues [22] showed that a 10-session RFT (duration of one session was 30 min) increased HRV among healthy individuals. In addition, two recently published studies indicate the potential of RFT to increase HRV in healthy individuals: Lin [7] in a four-week intervention study (one hour weekly at the laboratory and 10 min homework daily), and Schumann and colleagues [8] in an eight-week intervention study (five sessions; four sessions at home and one session at the laboratory). According to Steffen and colleagues [23], even short unique RFT sessions (15 min) can have effects on HRV. However, to date, none of the studies measured HRV daily in an ambulatory assessment for several weeks to investigate whether average RFT leads to gradual improvements of HRV.

### 1.2. Progressive Muscle Relaxation

Another possible method to improve autonomic and cardiovascular effects is progressive muscle relaxation (PMR), which teaches individuals to reduce their muscle tone [24]. Since muscles are part of the sympathetic nervous system the direct effect of PMR is to decrease the level of sympathetic arousal (thus the parasympathetic activity is indirectly increased), whereas RFT directly aims at strengthening the parasympathetic component [1]. Focusing on cross-sectional and pre- and post-measurement studies, Pawlow and Jones [25] demonstrated that PMR reduces blood pressure and heart rate (HR), but the measurement was carried out immediately after the PMR session, so the outlasted effect of the PMR cannot be analyzed. Indeed, Chaudhuri and colleagues [26] demonstrated this effect in highly stressed female health care professionals after practicing PMR for three months (participants were asked to complete 20 min of PMR on their own each day), but they did not analyze men, low and no stressed females, nor did they investigate the HRV. Seckendorff [27] examined this in his PhD thesis, indicating that a six-week training of PMR (6 × 60 min—under the guidance of a psychologist) did not have a significant effect on HR and HRV in healthy participants. Provided that the participants in the study of Chaudhuri and colleagues [26] practiced PMR every day, they underwent a longer training of PMR (about 1800 min of PMR) in terms of quantity compared to the participants in the study conducted by Seckendorff [27] (about 360 min of PMR). Maybe the PMR training in the study of Seckendorff [27] was too short (also in frequency) to find an effect on HRV. However, even that relatively short relaxation training (including PMR; one hour weekly in the laboratory and 10 min daily at home) can have positive effects on HRV was recently shown by Lin [7] in a four-week intervention. She also examined the effect of RFT and indicated that participants in the RFT group increased HRV to a slightly greater degree than participants in the relaxation group. However, again, none of the studies mentioned here measured HRV daily in an ambulatory assessment for several weeks to determine possible gradual improvements of HRV.

### 1.3. Present Study

Most previous studies, focusing on RFT and PMR, examined individuals who had autonomic nervous system (ANS) dysregulation (and consequently a compromised HRV), only analyzed the effects of relatively short-term interventions, or measured HRV only in cross-sectional or pre- and post-measurement designs. However, little can be said about the long-term effect of RFT and PMR in these studies, since HRV is on the one hand stable, but on the other hand also relatively variable (influenced by many environmental factors, e.g., physical, mental and health-related stressors; for reviews about influencing factors see [28,29]. For example, Bertsch and colleagues [30] indicate that 40% of HRV measurement variance was due to occasion-specific effects. Therefore, it seems useful to measure HRV in a relatively high frequency over a long period of time with ambulatory assessment to be able to find potential gradual effects. Ambulatory assessment in everyday life allows multiple measurements over a longer period of time within subjects. These intensive longitudinal data allow us to analyze the possible gradual effect of RFT and PMR within the subject more precisely than studies with pre- and post-measurements in the laboratory. Because no previous study has measured HRV with ambulatory assessment on a daily basis for several weeks, the present study is pilot in nature. More precisely, the present study tries to investigate the potential gradual long-term effects of average RFT and PMR over 77 days by daily measuring of HRV among healthy individuals with no obvious ANS dysregulation. The effects of average RFT and PMR were tested against an active control group, which completed a dual-task consisting of cognitive tasks (primarily focusing on working memory) and a motor task that was conducted at the same time. The control group was chosen so that at least small indirect effects on HRV would also have been possible, to strengthen the validity of the potential RFT and PMR effects. Due to the neurovisceral integration model [4], working memory capacity may be linked to HRV because of the common neural base—the prefrontal cortex. However, De Simoni and von Bastian [31] indicate that training interventions of working memory in young adults does not improve either working memory capacity or the working memory mechanisms that are supposed to underlie transfer. Therefore, the effect on HRV, if any, should be rather small, so that a significant gradual increase in HRV is expected for both the average RFT group and the PMR group compared to the active control group.

## 2. Methods

### 2.1. Participants

Participants were composed of 21 students (16 female, *M* = 21.24 years, *SD* = 1.57, range 19 to 26), so that each group, average RFT (seven females; *M* = 21.86 years, *SD* = 0.64, range 21 to 23), PMR (five females; *M* = 20.14 years, *SD* = 0.83, range 19 to 21), and dual-task (four females; *M* = 21.71 years, *SD* = 2.12, range 19 to 26) consisted of seven participants, respectively. The participants were randomly assigned to the groups by drawing a number.

### 2.2. Ethical Consideration

Before the experiment, participants received written informed consent, and we ensured them that their data would be treated confidentially. By participating in the study, the students could improve their grades in a seminar. However, they could also leave the study at any point in time and switch to a parallel seminar. The study was approved by the Ethics Committee of the university (02.2017).

### 2.3. Heart Rate Variability Measurement

For HRV measurement, we used the device and software of the VIITA HOLDING GmbH ((https://www.vital-monitor.com, Traun, Austria) accessed on 1 August 2021) with a breast belt (sampling rate of 500 Hz—1-Channel ECG), and a mobile phone app. One measurement took three minutes. The three-minute HRV measurement was divided into a spontaneous breathing phase and a short average RFT phase (i.e., each participant had a short average RFT, even though the participant was not part of the average RFT intervention group. This means that each participant has received a short average RFT training and a specific intervention). The spontaneous breathing phase took 110 s and the average RFT phase 70 s. During the measurement, which was supposed to be completed approximately 10 min after getting up in the morning, participants were asked not to speak or distract themselves. They were asked to sit on a chair with a backrest for each measurement so that minimal muscle tension was required to maintain the position. As HRV parameter, we used the root mean square of the successive differences (RMSSD), which was only calculated for the spontaneous breathing phase. Overall, 96 percent (range: 86–100 percent) of the daily measurements (77 days) were completed. The measurements were monitored daily, and in case of obvious measurement difficulties, communication with the participants was carried out via the chat function of the VIITA HOLDING GmbH software.

### 2.4. Intervention Groups and Active Control Group

#### 2.4.1. Average Resonance Frequency Training

Participants of the average RFT group were instructed in the first session to breathe nasally and at a frequency of about 0.1 Hz (5 s inhalation and 5 s exhalation) for five minutes on each of the following 77 days (385 min in total). Average RFT was performed and observed with the device and software of the VIITA HOLDING GmbH. To address difficulties and to see if participants were performing the tasks correctly, training was completed once a week with an experimenter. Overall, 99 percent (range: 99–100 percent) of the average RFTs were completed.

#### 2.4.2. Progressive Muscle Relaxation

Participants of the PMR group practiced PMR three times a week for approximately 18 min (overall approximately 594 min). For the training, we used an audio file (short version with background music; German version) of the “Techniker Krankenkasse” (available under www.tk.de/tk/gesunder-ruecken/entspannung-und-entlastung/jacobson/21538, accessed on 25 September 2016). Once a week, all participants of this group practiced together. The other two sessions were completed by the participants themselves and were observed and presented by the SoSci Panel, a service to create online questionnaires and studies (www.soscisurvey.com, accessed on 25 September 2016). The PMR training started with guided instruction on breathing. Participants were then guided to contract the muscles in their right and left hands, as well as their arms, and were then asked to release tension after a few seconds. Participants were guided to repeat this contraction and relaxation ritual for three other muscle groups (the head, the trunk, and the right and left upper legs, calves, and feet). This was followed by some minutes of background music, where participants were invited to imagine a beautiful place of rest (i.e., resting image). Overall, 99 percent (range: 94–100 percent) of the PMR training was completed.

#### 2.4.3. Dual-Task for the Active Control Group

The dual-task consisted of four different cognitive tasks (primarily focusing on working memory), which were combined with a simultaneous motor task (participants had to keep their balance on a Moonhopper). The cognitive tasks were presented by a video, and participants practiced three times a week for 20 min, respectively (overall: 660 min). Once a week, all participants practiced together, to address difficulties and to see if participants were performing the tasks correctly. The remaining sessions were completed by the participants themselves and were observed and presented by the SoSci Panel. Overall, 93 percent (range: 76–100 percent) of the dual-tasks were done.

### 2.5. Data Analyzing and Processing

After downloading the RR intervals from the VIITA HOLDING GmbH server, we analyzed the data with the Kubios HRV analysis package (version 3.3.1, Kuopio, Finland) [32] to calculate the RMSSD. For HRV analysis, the first 10 s of each measurement was automatically removed by the VIITA HOLDING GmbH software, so that in total, 100 s of the spontaneous breathing condition were analyzed. Because the software only records IBIs, we used the automatic filter (threshold level: very low) offered by Kubios to detect RR intervals that differ “abnormally” from the normal mean RR interval, which may represent an artifact. This procedure is commonly used, and in most cases considered sufficient, when data were recorded in rest conditions [33]. The calculated RMSSD values were ln transformed because of the non-normal distribution.

This study was supposed to last 14 weeks, but due to travel (i.e., jetlag), longer periods of illnesses, and stress caused by the examination phase at the end of the semester, our study had to be reduced to 11 weeks.

A mixed-effect model with an autoregressive error term (AR1), using random slopes (daily HRV measurement—these values were *z*-standardized for analysis), and random intercepts (participants) was calculated using the glmmTMB package [34] and the R software (version 4.0.5, www.r-project.org, Vienna, Austria (accessed on 1 August 2021)). For the AR1 the daily HRV measurement was used as factor and the preliminary code was as follows:(1)ln RMSSD ~ Day∗Group+(1+Day | Participant)+AR1(factor(Day)−1 | Participant)

Since we used both random slopes and random intercepts for the analysis, the control of confounding variables was not as crucial for the interaction term of daily HRV measurement and group. However, due to the small sample size and since we also wanted to test if the average RFT and the PMR group differ in HRV from the active control group, we controlled for age (HRV decreases with age [35,36]), sex (even though Koenig & Thayer [37] have shown that men and women do not differ on the ln RMSSD parameter), BMI (body mass index; obese people have lower HRV [38]) and positive and negative trait affect regulation (HRV and positive affect are associated in an upward spiral, meaning that improved positive affect regulation is reciprocally associated with enhanced HRV [39,40,41,42]). To assess individual differences in positive and negative trait affect regulation, participants completed the German version of the Action Control Scale (ACS-90, [43]). Here, positive trait affect regulation can be assessed with the demand-related action-state orientation (AOD/SOD) subscale whereas negative trait affect regulation can be assessed with the failure-related action-state orientation (AOF/SOF). After adding the control variables, the code of the mixed-effect model was as follows:(2)lnRMSSD ~ Day∗Group+Age+Sex+BMI+AOD/SOD+AOF/SOF+(1+Day | Participant)+AR1(factor(Day)−1 | Participant)

## 3. Results

Descriptive statistics for raw RMSSD are shown in Table 1.

First of all, the mixed-effect model (see Table 2) with AR1, using random slopes (daily HRV measurement) and random intercepts (participants), indicated no significant main effect of the control variables. Focusing on the effects of RMR and RFT the results indicated that the PMR group significantly increased their HRV, compared to the active control group; *β* = 0.10, *p* = 0.028 (see Figure 1). For the average RFT group, an increase in HRV could not be observed; *β* = 0.01, *p* = 0.911 (see Figure 1), compared to the active control group. To illustrate the results more precisely, Figure 2 shows the HRV changes for each participant individually.

## 4. Discussion

This pilot study tested whether gradual HRV improvements are possible in healthy individuals with no obvious ANS dysregulation through average RFT and PMR, when HRV was measured daily in the morning with ambulatory assessment. Such intensive intervention studies with frequent measurements are rather rare, because in previous intervention studies HRV was often measured only twice—at the beginning and at the end of interventions. As far as we know, this is the first intervention study that measured HRV in healthy individuals daily over a longer period, investigating the long-term gradual effects of PMR and RFT. The result indicated a significant gradual increase in HRV (ln RMSSD) in the PMR group, but not for the average RFT group, compared to the active control group, respectively. Therefore, this pilot study indicates that frequent and long-term PMR training has the potential to gradually increase HRV among healthy individuals. Here, the result seems to contradict the findings of Seckendorff [27] regarding PMR (6 x 60 min) who found no HRV improvement, but is in line with Lin [7] who indicated that relatively short relaxation training (including PMR; one hour weekly in the laboratory and 10 min daily at home) had positive effects on HRV. However, Lin [7] also analyzed the effect of RFT and showed that RFT increased HRV even to a slightly larger degree than relaxation training. In contrast, the daily average RFT in our pilot study did not lead to a gradual improvement in HRV. This might be the case since the average RFT group had a slightly higher but not significant general HRV than the active control group. The possible explanation of higher HRV is in line with the results of Schumann and colleagues [8], who indicated that an eight-week RFT training led to an increase in HRV, but that the training had no effect for individuals with higher HRV. Therefore, it seems that especially individuals with low HRV benefit from (average) RFT. Another possible explanation for this non-significant result could be that only an average RFT was used. The individual resonance frequency breathing rate, typically ranging between 4.5 and 6.5 breaths/minute, can be identified by asking a person to breathe at 4.5, 5.0, 5.5, 6.0 and 6.5 breaths/minute during HRV recording to find the breathing rate that results in the largest changes in HRV (where HRV becomes maximum, [11,12,22]). Subsequent studies should consider this possibility in more detail.

What might also play a role in the effectiveness of interventions is that it is not clear today how much training is needed to gradually increase the HRV (i.e., the dose-effect relation). In this pilot study, participants in the average RFT group breathed for five minutes daily. This might be too short an intervention regarding the duration of a single session (in most studies, single breathing sessions are often much longer than 5 min), even if the total duration of the intervention (385 min) was rather high. In the PMR group the participants had longer single sessions (about 18 min per session), and also the total duration of the intervention was longer (594 min) than in the average RFT group, both of which could potentially be reasons why there was an improvement in the PMR group and not in the average RFT group.

One further possibility why there was no gradual increase in HRV in the average RFT group could also be that the participants did not breathe in the given respiratory frequency (6 breaths/minute). The general performance of the RFT was monitored with the VIITA HOLDING GmbH software, but not the respiratory frequency itself. In further studies, the respiratory frequency of participants in RFT should also be monitored using a respiratory belt or by calculating the respiratory rate from the ECG signal (e.g., using the Kubios respiratory algorithm [32]) to verify that participants are actually breathing at the specified frequency.

### Some Further Proposals for Subsequent Studies

Our pilot study represents a relatively small sample, where individuals measured their HRV relatively often. Subsequent studies should validate our findings using a more balanced sex ratio and a larger sample size. Moreover, further studies should also investigate whether the results can be transferred to other populations outside the student population (e.g., elderly people).

Focusing on the effect of PMR, it should be noted that perhaps not only the PMR training alone improved the HRV, but also the combination of PMR and the 70 s of average RFT in the daily HRV measurement. This possibility should be excluded in future studies.

Last but not least, further studies should also test whether a gradual HRV improvement is also reflected in healthy individuals in the other energy system, the hypothalamic-pituitary-adrenal (HPA) axis [44], or in better self-regulation at the emotional, cognitive, social, and health domains [2,3,4]. For example, Schumann and colleagues [8] had recently shown that RFT can increase HRV in healthy individuals but that this increase had no effect on measures of behavioral impulsivity. However, further intensive intervention studies are certainly needed to get a clearer picture of whether HRV interventions also affect other variables.

## 5. Conclusions

This pilot study tested whether gradual HRV improvements are possible in healthy individuals with no obvious ANS dysregulation through average RFT and PMR, when HRV was measured daily in the morning with ambulatory assessment over 77 days. Such intensive intervention studies with frequent measurements are rather rare, because in previous intervention studies HRV was often measured only twice—at the beginning and at the end of an intervention. The results indicated that the PMR group significantly increased their HRV compared to the active control group. However, no effect was observed for the RFT group. Here, further studies considering the limitations of this pilot study are needed to validate the results.

## Figures and Tables

**Figure 1 ijerph-18-11357-f001:**
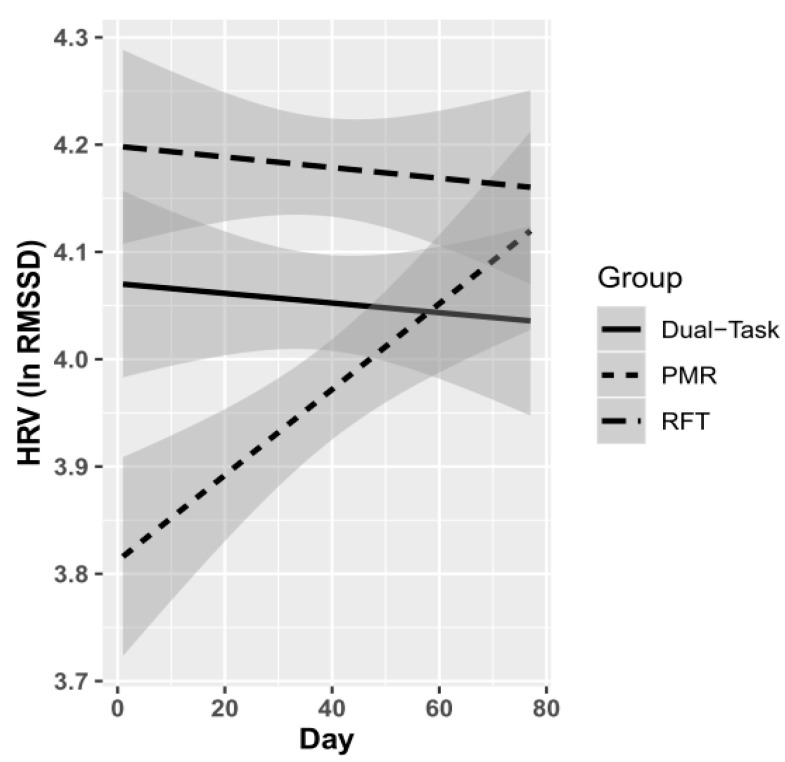
Time-varying effects of each group (dual-task = control group; PMR = progressive muscle relaxation; RFT = average resonance frequency training); HRV = heart rate variability; Day = daily HRV measurement; Grey shades correspond to the 95% confidence interval, respectively.

**Figure 2 ijerph-18-11357-f002:**
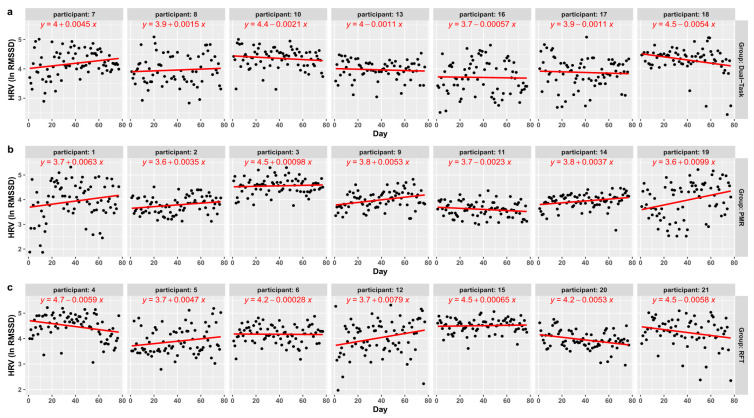
(**a**–**c**) Daily HRV measurement of each participant; the graph also includes the individual linear equation (random effects) of each participant.

**Table 1 ijerph-18-11357-t001:** Descriptive statistics of raw RMSSD.

	Dual-Task	PMR	RFT
*M*	64.65	61.22	74.21
*SD*	29.62	33.95	36.00
max	162.34	204.52	203.71
min	11.50	6.47	7.19
*n* (measurements)	501	531	522

*Note.* Dual-Task = control group; PMR = progressive muscle relaxation; RFT = average resonance frequency training.

**Table 2 ijerph-18-11357-t002:** Linear Mixed Effect Model, using Resting HRV (ln RMSSD) as the Criterion.

Predictors	Estimates	CI	*p*
(Intercept)	4.63	2.36–6.90	<0.001
Day	−0.02	−0.08–0.05	0.626
PMR	−0.24	−0.55–0.07	0.126
RFT	0.14	−0.21–0.48	0.433
Age	−0.06	−0.14–0.03	0.204
Sex (men)	0.01	−0.52–0.55	0.959
BMI	0.03	−0.03–0.09	0.301
AOD/SOD	−0.01	−0.05–0.03	0.596
AOF/SOF	0.02	−0.04–0.07	0.619
Day × PMR	0.10	0.01–0.20	0.028
Day × RFT	0.01	−0.09–0.10	0.911
**Random Effects**			
	**Variance**	**Standard Deviation**	**Correlation**
Participant	0.0319	0.1787	
Day	0.0010	0.0315	−0.15
AR1	0.0546	0.2336	0.77
Residual	0.1737	0.4168	
	**AIC**	**BIC**	**Log Likelihood**
Model Statistics	2103.8	2194.7	−1034.9

*Note.* Observations: 1554; Results were tested against the active control group; HRV = heart rate variability; Day = daily HRV measurement; PMR = progressive muscle relaxation; RFT = average resonance frequency training; BMI = body mass index; AOD/SOD = trait positive affect regulation; AOF/SOF = trait negative affect regulation; AR1 = autoregressive error term; x = interaction.

## Data Availability

Data can be shared by the corresponding author upon reasonable request.

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
