# Peer review of "Increasing Heart Rate Variability through Progressive Muscle Relaxation and Breathing: A 77-Day Pilot Study with Daily Ambulatory Assessment"

_ijerph, 2021, doi:10.3390/ijerph182111357_

Round 1
Reviewer 1 Report
The study of Grob et al (1365783) investigated whether it is possible to gradually increase HRV in response to regular exercises of average resonance frequency training and progressive muscle relaxation. The authors reported a significant gradual increase in HRV (ln RMSSD) in the PMR group, but not for the average RFT group, compared to the active control group, respectively. The study is well designed and nicely executed. There are, however, few data interpretational issues that authors need to clarify and subsequently address in their manuscript.
Major Points
- The rationale of this study is not very clear.
- A testable hypothesis is not presented.
- The practical considerations are not highlighted.
- The description of the methods is not so clear.
Specific Points
The following points need to be addressed:
Introduction:
The rationale of this study is not very clear. A testable hypothesis is not presented. What are the practical considerations of this research?
The introduction section is too long.
Paragraph 1, lines 28-31: It is often demonstrated in clinical….post-measurements design’ It does not make sense, the authors should mention the reason for HRV increase or decrease in the clinical and healthy population and whether these changes are due to a specific stimulus. Please rewrite it.
Method.
Participants
- The majority of the participants were female (16/21). Please provide information about whether the phase of the menstrual cycle was controlled before and after the exercise training program evaluations and indicated the phase of the menstrual cycle. The cyclic variations of estrogen and progesterone levels during a menstrual cycle could affect the cardiovascular regulation
Discussion
It is well written without raising major concerns.
Author Response
Response to Reviewer 1 Comments
The study of Groß et al (1365783) investigated whether it is possible to gradually increase HRV in response to regular exercises of average resonance frequency training and progressive muscle relaxation. The authors reported a significant gradual increase in HRV (ln RMSSD) in the PMR group, but not for the average RFT group, compared to the active control group, respectively. The study is well designed and nicely executed. There are, however, few data interpretational issues that authors need to clarify and subsequently address in their manuscript.
Major Points
- The rationale of this study is not very clear.
- A testable hypothesis is not presented.
- The practical considerations are not highlighted.
- The description of the methods is not so clear.
Specific Points
The following points need to be addressed:
Introduction:
The rationale of this study is not very clear. A testable hypothesis is not presented. What are the practical considerations of this research?
The introduction section is too long.
Paragraph 1, lines 28-31: It is often demonstrated in clinical….post-measurements design’ It does not make sense, the authors should mention the reason for HRV increase or decrease in the clinical and healthy population and whether these changes are due to a specific stimulus. Please rewrite it.
Response 1: Thank you for the hints. We have rewritten and shorten the introduction in some parts (see manuscript) in order to give a better rational for the study. The practical considerations of the study are whether RFT and PMR have a long-term effect on HRV – visible by a gradual increase of HRV. Therefore, we formulated the following hypothesis “…, so that a significant gradual increase in HRV is expected for both the average RFT group and the PMR group compared to the active control group.”, which is presented in the last sentence of the introduction. Also, in the following sentence, we have given the reason why HRV increase has occurred (slow breathing). “It is often demonstrated in clinical populations (however, in this population HRV is often greatly reduced), that HRV can be increased through slow breathing, but also some studies indicated HRV increase for healthy individuals in longitudinal studies using a pre- and post-measurement design (e.g., Gevirtz, 2013; Lin, 2018; Schumann et al., 2019).”
Method.
Participants
- The majority of the participants were female (16/21). Please provide information about whether the phase of the menstrual cycle was controlled before and after the exercise training program evaluations and indicated the phase of the menstrual cycle. The cyclic variations of estrogen and progesterone levels during a menstrual cycle could affect the cardiovascular regulation
Response 2: It is true that the menstrual cycle has an effect on HRV, as many other variables have an effect on HRV. We did not specifically control for the menstrual cycle because our interest was in the gradual increase in HRV over 77 days, which should be independent of the menstrual cycle. Since we used both random slopes and random intercepts for the analysis, the control of confounding variables was not as crucial for the interaction term of daily HRV measurement and group.
Discussion
It is well written without raising major concerns.

Reviewer 2 Report
Dear Authors, the article “Increasing heart rate variability through progressive muscle relaxation and breathing: A 77-day pilot study with daily ambulatory assessment” is interesting. This pilot study tested whether gradual HRV improvements are possible in healthy individuals with no obvious ANS dysregulation through average RFT and PMR, when HRV was measured daily in the morning with ambulatory assessment. But several parts of the manuscript could be corrected.
General comments:
- I recommend explaining why the classical control group was not used and why did you use this kind of control group.
- In conclusion, I lack a summary of the benefits of the research.
- Did you run an a priori power analysis? It concerns the number of participants required.
Specific comments:
- Line 30 – change incease to increase
- Line 39-40: You might check updated studies. There are studies with daily HRV measurements over several weeks, e.g. guided training at altitude.
- In Introduction section - Try expand the HRV section.
- Line 32-33: I recommend that the study's objectives and tasks be placed at the end of the introduction section.
- What methodology did the participants use to check their respiratory rate? You probably have not recorded information about respiratory frequency, which is the important limit of a study in terms of results. This information would greatly improve research.
- References: Several studies on the use of HRV in sport and exercise activities have been published over the past 2 years.
Author Response
Response to Reviewer 2 Comments
Dear Authors, the article “Increasing heart rate variability through progressive muscle relaxation and breathing: A 77-day pilot study with daily ambulatory assessment” is interesting. This pilot study tested whether gradual HRV improvements are possible in healthy individuals with no obvious ANS dysregulation through average RFT and PMR, when HRV was measured daily in the morning with ambulatory assessment. But several parts of the manuscript could be corrected.
General comments:
- I recommend explaining why the classical control group was not used and why did you use this kind of control group.
Response 1: Thank you for this hint. At the end of the introduction we explained why we used this type of control group. “The effects of average RFT and PMR were tested against an active control group, which did a dual-task consisting of cognitive tasks (primarily focusing on working memory) and a motor task that was conducted at the same time. The control group was chosen so that at least small indirect effects from it on HRV would also have been possible, to strengthen the validity of the potential RFT and PMR effects. Due to the neurovisceral integration model (e.g. Thayer et al., 2009), working memory capacity may be linked to HRV because of the common neural base – the prefrontal cortex. However, De Simoni and von Bastian (2018) indicate that training interventions of working memory in young adults does not improve either working memory capacity or the working memory mechanisms that are supposed to underlie transfer. Therefore, the effect on HRV, if any, should be rather small, so that a significant gradual increase in HRV is expected for both the average RFT group and the PMR group compared to the active control group.”
- In conclusion, I lack a summary of the benefits of the research.
Response 2: Thank you for this hint. We pointed out the specific benefit of the study at the very beginning of the discussion, stating that: “Such intensive intervention studies with frequent measurements are rather rare, because in previous intervention studies HRV was often measured only twice - at the beginning and at the end of an intervention. As far as we know, this is the first intervention study that measured HRV in healthy individuals daily over a longer period investigating long-term gradual effects of PMR and RFT.”
- Did you run an a priori power analysis? It concerns the number of participants required.
Response 3: No, we did not run a power analysis. Since there is no longitudinal study yet that has examined the gradual effect of PMR and RFT on HRV through daily ambulatory assessment, this study is a pilot study. Also, it is not possible to reliable determine a priori the number of participants in the statistical method we used without previous studies. We used a mixed-effect model with an autoregressive error term (AR1), using random slopes (daily HRV measurement) and random intercepts (participants)
Specific comments:
- Line 30 – change incease to increase
Response 4: Thank you for this hint. We changed the typo.
- Line 39-40: You might check updated studies. There are studies with daily HRV measurements over several weeks, e.g. guided training at altitude.
Response 5: That's right. The sentence must be seen from the background of RFT and PMR. Nevertheless, the sentence is no longer found in the revised manuscript.
- In Introduction section - Try expand the HRV section.
Response 6: In the revised manuscript, we have tried to expand the HRV section somewhat, but always with a focus on not making the introduction too long (see Reviewer 1).
- Line 32-33: I recommend that the study's objectives and tasks be placed at the end of the introduction section.
Response 7: In the revised manuscript, the objectives are now only at the end of the introduction section. Thank you for this idea.
- What methodology did the participants use to check their respiratory rate? You probably have not recorded information about respiratory frequency, which is the important limit of a study in terms of results. This information would greatly improve research.
Response 8: This is a good point. We did not check the respiratory frequency and reported this limitation in the discussion as the following: “One further possibility why there was no gradual increase in HRV in the average RFT group could also be that the participants did not breathe in the given respiratory frequency (6 breaths per minute). The general performance of the RFT was monitored with the VIITA HOLDING GmbH software, but not the respiratory frequency itself. In further studies, the respiratory frequency of participants in RFT should also be monitored using a respiratory belt or by calculating the respiratory rate from the ECG signal (e.g., using the Kubios respiratory algorithm [Tarvainen et al., 2014]) to verify that participants are actually breathing at the specified frequency (in our case 6 breaths per minute).”
Checking the respiratory frequency should be done in later studies, although it also cannot be verified that subjects are actually performing PMR right. Verification of correct performance always remains the difficulty of some interventions.
- References: Several studies on the use of HRV in sport and exercise activities have been published over the past 2 years.
Response 9: Thank you for this comment. In our study, we wanted to focus only on studies that focused on slow breathing and PMR to increase HRV. Therefore, we do not have integrated other studies (see also Response 5).

Round 2
Reviewer 2 Report
Dear authors,
my questions are answered, congratulations on the research.